# Low growth resilience to drought is related to future mortality risk in trees

Lucía DeSoto [iD] et al.#

Severe droughts have the potential to reduce forest productivity and trigger tree mortality. Most trees face several drought events during their life and therefore resilience to dry conditions may be crucial to long-term survival. We assessed how growth resilience to severe droughts, including its components resistance and recovery, is related to the ability to survive future droughts by using a tree-ring database of surviving and now-dead trees from 118 sites (22 species, >3,500 trees). We found that, across the variety of regions and species sampled, trees that died during water shortages were less resilient to previous non-lethal droughts, relative to coexisting surviving trees of the same species. In angiosperms, drought-related mortality risk is associated with lower resistance (low capacity to reduce impact of the initial drought), while it is related to reduced recovery (low capacity to attain pre-drought growth rates) in gymnosperms. The different resilience strategies in these two taxonomic groups open new avenues to improve our understanding and prediction of drought-induced mortality.

---

#A full list of authors and their affiliations appears at the end of the paper.

Forests provide essential ecosystem services[1–4], yet are strongly threatened by deforestation, fragmentation and climate change[5,6]. Particularly, drought events associated with increasing temperatures have the potential to reduce forest productivity and prompt tree mortality in many areas of the world[7–10]. Models of climate change predict significant increases in the frequency, duration and severity of droughts in vast regions of the globe[11]. Alleviating negative effects of climate change on forests requires global and long-term strategies[12]. Therefore, there is an urgent need to better understand the processes underlying drought-induced tree mortality worldwide as a prerequisite to adapt forest management strategies to climate change.

Many studies have assessed the physiological mechanisms related to carbon and water economy, underlying drought-induced mortality with the aim of developing reliable, mechanistic indicators of mortality risk (e.g. refs. [13–16]). Other efforts have been directed towards more empirical indicators, usually based on radial growth as a compound measure of tree vitality (e.g. refs. [17–22]). An important factor to consider is that, while severe drought may trigger tree mortality within a population, some trees can be less vulnerable to dry conditions than others and survive[14,23–25]. Since most trees face several drought events during their lives, high resilience to drought might determine long-term tree survival.

Resilience describes the capacity of a system to maintain its functions after the impact of an exogenous disturbance[26]. Some studies suggested that low resilience to drought may boost tree mortality risk[27–29]. However, to our knowledge, no study has evaluated the direct linkage between resilience to drought and future mortality risk mainly because it is difficult to empirically evaluate both resilience and mortality on the same individual tree.

Short- and long-term responses of trees to drought can be assessed using tree-ring data, allowing a retrospective quantification of drought effects at annual resolution for numerous individuals, sites and species (e.g. refs. [30,31]). Growth resilience can be defined as the capacity of a tree to reach growth rates similar to those prior to drought. Defined this way, resilience encompasses the capacity to reduce the impact of the disturbance, i.e. resistance, and the ability to return to pre-disturbance growth levels after drought, i.e. recovery[32,33]. These two components of resilience may vary within taxonomic groups. Pinaceae species (gymnosperms) tend to show stronger and longer legacies in radial growth after drought (slower recovery) than Fagaceae species (angiosperms)[30,34]. Nonetheless, little is known about how these legacies affect the ability of trees to cope with future drought events in terms of mortality risk, which would determine long-term demographic responses.

Here we took advantage of a recently assembled pancontinental database[22] to study the relationship between past resilience to drought and mortality risk under subsequent drought events. The database contains tree-ring width (TRW) series for surviving and dead trees from 118 sites and >3500 individuals around the globe (mostly from the temperate, Mediterranean and boreal ecosystems of the Northern hemisphere), including 22 species (8 angiosperms and 14 gymnosperms; Fig. 1, Supplementary Data 1). Because surviving and dead trees were sampled concurrently at the same sites (matched-pairs case–control study), this database is ideal to assess and compare growth patterns of trees before mortality[22]. We quantified tree resilience to drought using three indices that were proposed by Lloret et al.[33] and calculated them on time series of both TRW and basal area increment (BAI). The indices are: (1) resistance, the ratio between radial growth during the drought year and radial growth in the period immediately before; (2) recovery, the ratio between radial growth in the period immediately after the drought and radial growth during the drought year; and (3) resilience, the ratio between post-drought and pre-drought radial growth. Based on mixed-effect models, we then analysed the relationships between these three resilience indices and future mortality risk, also accounting for the effects of taxonomic group (angiosperms vs. gymnosperms) and several variables characterising environmental conditions and tree size (see "Methods").

We hypothesised that trees that ultimately died during drought events (hereafter now-dead trees) were already less resilient to droughts that occurred decades before their death, relative to surviving trees from the same population. We also expected that the nature of this relationship would differ between angiosperm and gymnosperm trees, due to their contrasting trait syndromes and drought response strategies[30,34,35]. Finally, we assessed how the relationship between drought resilience and future mortality risk depends on (i) the long-term water availability of each site (characterised by the aridity index calculated as the ratio between precipitation and potential evapotranspiration (PET)), (ii) the intensity of the drought event under consideration and (iii) soil properties.

Our findings confirm that trees that died during water shortages were less resilient to previous non-lethal droughts, relative to coexisting surviving trees of the same species. This is,

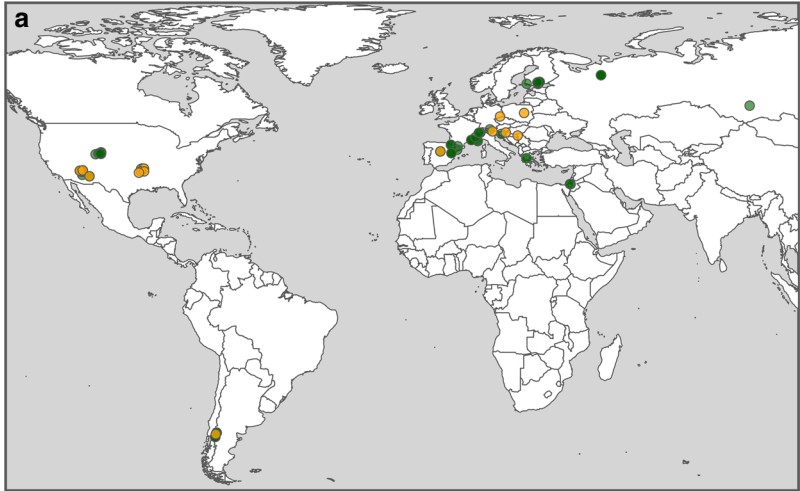
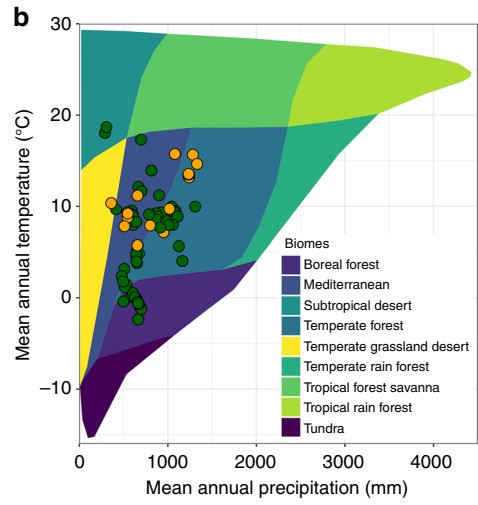

**Fig. 1 Spatial and climatic ranges of the study. a** Geographical distribution and **b** Whittaker biome classification for the study sites. The angiosperm (orange) and gymnosperm (green) tree species included in the analysis are depicted (see Supplementary Data 1 for the description of the populations).

to our knowledge, the first empirical evidence linking low growth resilience to past droughts with increased mortality risk across tree species and regions. Although this result is consistent for angiosperms and gymnosperms, we show that the key component of resilience that is involved (resistance vs. recovery) differs between these two taxonomic groups. This is in accordance with the fundamental differences in their drought response strategies, such as avoidance of hydraulic failure or allocation to carbon storage.

## Results

**Mortality risk is associated with low resilience to drought.** Our results show that trees that died because of drought were indeed less resilient to previous droughts occurring decades before their death, relative to coexisting surviving trees of the same species (Fig. 2a). This pattern is observed across the variety of regions we studied and for most of the species we sampled (Fig. 1 and Supplementary Fig. 4) and it is consistent for both gymnosperms and angiosperms (Table 1 and Supplementary Table 1). Within sites, differences in drought resilience between coexisting trees may be caused by differences in micro-environmental conditions, such as competition and intra-plot heterogeneity in soil properties (extrinsic), or in traits that determine the plant water and carbon economies (intrinsic)[14,36,37]. When xylem growth rates (and consequently tree-ring formation) decrease as a result of drought, water transport as well as carbon availability are compromised, potentially affecting subsequent growth[38]. This is because xylem is the responsible tissue for long-distance transport of sap from roots to leaves and for storage of large amounts of carbohydrates. Drought-related reductions in radial growth may not only reflect direct (deleterious) effects on turgor-driven cell growth or carbon availability but may also cause structural adjustments, such as plastic changes in resource allocation because of dry conditions[37]. These two effects are difficult to disentangle under field conditions. However, the fact that sustained declines of TRW are associated with increased mortality risk[22] coupled with our finding of lower resilience linked to higher mortality risk indicates that deleterious effects probably might be dominant in this case. Therefore, the relationship between resilience in xylem growth to past droughts and future susceptibility to drought may provide a valuable and generalisable proxy for future mortality risk assessment at the individual tree level.

**Angiosperms and gymnosperms differ in their strategies.** We find that the absolute values of resilience, as well as the overall relationship between resilience and future mortality risk, are similar for angiosperms and gymnosperms (Table 1). This is noteworthy, considering the well-known differences in drought resilience strategies and related traits between these two taxonomic groups[15,35,39,40]. However, the specific associations between survival and the components of resilience (resistance and recovery) differ between angiosperms and gymnosperms (Table 1, Fig. 2b, c). Surviving angiosperm trees were more resistant to droughts occurring some decades before but show similar recovery capacity compared to now-dead trees (Fig. 2b). Surviving gymnosperm trees also show slightly increased resistance, but the main difference between surviving and now-dead gymnosperms is that surviving trees recovered better from previous droughts (Fig. 2c). These patterns are consistent regardless of the response variable used to characterise resilience (TRW or BAI; Supplementary Tables 3 and 4; Supplementary Fig. 2). Limited resistance implies low capacity to reduce the impact of the drought in now-dead angiosperms, whereas limited recovery

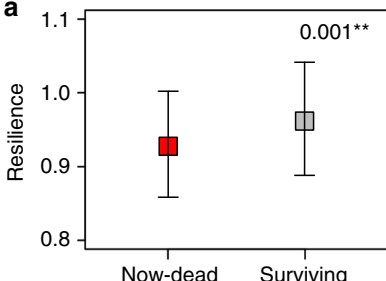

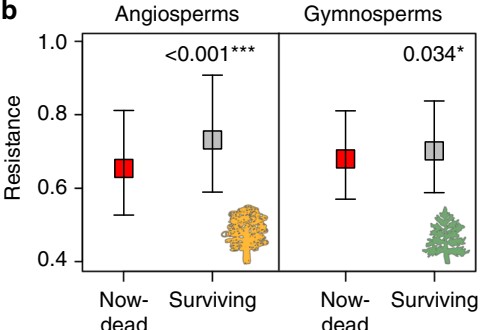

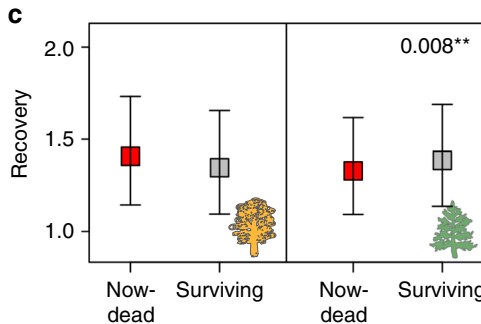

**Fig. 2 Differences in resilience, resistance and recovery between now-dead and surviving trees. a** Differences in resilience between now-dead and surviving trees. Differences in **b** resistance and **c** recovery between now-dead and surviving trees as a function of the taxonomic group (angiosperms vs. gymnosperms). The data are presented as model-adjusted, back-transformed least-square means ± 95% confidence intervals (Table 1). Resilience, resistance and recovery indices were computed from tree-ring width (TRW) series of surviving (grey squares) and now-dead (red squares) trees. Asterisks indicate significant pairwise differences in least square means between now-dead and surviving trees ($t$ or $\chi^2$ test in LMM: *$P < 0.05$; **$P < 0.01$; ***$P < 0.001$). Panels are separated by taxonomic group only when differences between angiosperms and gymnosperms are significant (Table 1). Source data are available in Digital. CSIC repository (https://doi.org/10.20350/digitalCSIC/10536).

entails reduced ability to return to the pre-drought state in now-dead gymnosperms (Fig. 3).

Previous research suggests that hydraulic failure alone can usually explain drought-induced mortality in angiosperms, whereas in gymnosperms the carbon economy may also be involved[15]. This is consistent with narrower hydraulic safety margins (i.e., the buffer between minimum water potential experienced by the tree in the field and the empirical threshold of water potential for rapid loss of vascular function caused by embolism) in angiosperms compared to gymnosperms[39,40]. However, this difference must be interpreted with caution due to the difference in the critical percentage loss of hydraulic conductivity causing mortality in these two groups[41]. Despite the

**Table 1 Summary of the fitted linear mixed model of resilience, resistance and recovery.**

| | Std. $\beta$ | CI | df | $t/\chi^2$ | P |
|---|---|---|---|---|---|
| **Resilience** | | | | | |
| Fixed effects | | | | | |
| (Intercept) | **−0.062** | **−0.121, −0.004** | **72.9** | **−2.00** | **0.049** |
| Surviving | **−0.091** | **−0.142, −0.037** | **2623.9** | **−3.43** | **0.001** |
| DBH$_i$ | **−0.001** | **−0.001, 0.000** | **2536.1** | **−4.04** | **<0.001** |
| Δtime | **0.001** | **0.000, 0.002** | **1920** | **2.48** | **0.013** |
| Aridity | 0.011 | −0.056, 0.076 | 36.7 | 0.28 | 0.783 |
| Soil fertility | −0.002 | −0.011, 0.008 | 26.7 | −0.39 | 0.698 |
| Surviving × Δtime | **0.001** | **0.000, 0.002** | **3008.7** | **2.83** | **0.005** |
| Surviving × aridity | **0.063** | **0.027, 0.097** | **3143.6** | **3.51** | **<0.001** |
| Surviving × soil fertility | **−0.015** | **−0.021, −0.008** | **2679.8** | **−4.63** | **<0.001** |
| Random effects | | | | | |
| Genus (species (site)) | | | 3 | 739 | <0.001 |
| No. of trees/sites/species/genus | 3207/104/21/10 | | | | |
| $R^2$m/$R^2$c/ΔAIC | 0.06/0.3/22.2 | | | | |
| **Resistance** | | | | | |
| Fixed effects | | | | | |
| (Intercept) | **−0.207** | **−0.317, −0.077** | **36.9** | **−3.41** | **0.002** |
| Surviving | 0.026 | −0.009, 0.061 | 3557.8 | 1.47 | 0.143 |
| Gymnosperms | 0.017 | −0.066, 0.100 | 19.7 | 0.37 | 0.714 |
| Aridity | 0.037 | −0.102, 0.146 | 34.3 | 0.63 | 0.531 |
| Soil fertility | **0.016** | **0.001, 0.036** | **28.6** | **2.16** | **0.039** |
| Surviving × gymnosperms | **−0.035** | **−0.061, −0.009** | **3551.6** | **−2.66** | **0.008** |
| Surviving × aridity | 0.028 | −0.008, 0.065 | 3565.1 | 1.52 | 0.129 |
| Random effects | | | | | |
| Genus (species (site)) | | | 3 | 1293 | <0.001 |
| No. of trees/sites/species/genus | 3660/118/22/10 | | | | |
| $R^2$m/$R^2$c/ΔAIC | 0.04/0.47/18.3 | | | | |
| **Recovery** | | | | | |
| Fixed effects | | | | | |
| (Intercept) | **0.141** | **0.066, 0.216** | **29.4** | **3.56** | **0.001** |
| Surviving | −0.022 | −0.047, 0.002 | 3623.8 | −1.78 | 0.074 |
| Gymnosperms | −0.025 | −0.109, 0.059 | 22.1 | −0.56 | 0.578 |
| Soil fertility | **−0.016** | **−0.031, −0.003** | **34.4** | **−2.40** | **0.022** |
| Surviving × gymnosperms | **0.037** | **0.0100, 0.065** | **3628.1** | **2.65** | **0.008** |
| Surviving × soil fertility | **−0.006** | **−0.011, 0.000** | **3637.4** | **−2.07** | **0.039** |
| Random effects | | | | | |
| Genus (species (site)) | | | 3 | 1092 | <0.001 |
| No. of trees/sites/species/genus | 3733/118/22/10 | | | | |
| $R^2$m/$R^2$c/ΔAIC | 0.04/0.42/4.6 | | | | |

The response variables are log-transformed resistance, recovery and resilience computed for tree-ring width (TRW) data, assuming a Gaussian error distribution with an identity link. The fixed part of the model included status (now-dead or surviving), taxonomic group (angiosperm or gymnosperm), diameter at breast height (DBH$_i$, cm), time period between drought event and last year recorded in each individual tree ring-width series (Δtime, years), average ratio between precipitation and potential evapotranspiration (aridity) for the period 1970–2000, a measure of soil fertility, and interactions between status and other fixed effects. The random part of the model included site nested within species nested within genus. The intercept corresponds to the reference status (now-dead) and taxonomic group (angiosperms). This summary corresponds to the reduced model (the full model is presented in Supplementary Table 1; for model selection, see Supplementary Table 2). Values represent the standardised estimates of regression coefficients (std. $\beta$), 95% confidence intervals (CIs), the $t$ statistic or $\chi^2$ statistic and the associated $P$ value of significance (bold type for significant fixed effects, $P < 0.05$). Estimates of regression coefficients for the intercept were not standardised. The signs indicate the direction of the effects. $R^2$m is the marginal $R^2$, $R^2$c is the conditional $R^2$, ΔAIC is the increment on AIC values with respect to that of the model without status (Supplementary Table 2). The low marginal $R^2$ explained by the fixed effects of the reduced models might be a consequence of data heterogeneity, with high variation within species and sites[22]. Nevertheless, differences between statuses were detected, and smaller AICs and larger differences (ΔAIC) > 2.0 related to models without status indicate that models including status showed higher explanatory power[69].

high recovery capacity of angiosperms, differences in their ability to resist previous droughts are carried over and eventually result in either death or survival when trees are exposed to an even more severe subsequent drought.

In gymnosperms, negative effects on growth may persist beyond 4 years after the drought in now-dead trees, implying long-term legacy effects[22,30]. The gymnosperms that recovered less are more prone to die one or more decades after the severe droughts we studied (Fig. 3b, Supplementary Fig. 3). The greater importance of recovery in gymnosperms is consistent with the fact that their internal carbon reserves may be severely impacted under extreme drought[15]. Recovery of gymnosperms from drought may be compromised by their low capacity to refill embolized xylem tracheids[42], by their small fraction of

parenchyma tissue associated with lower levels of carbohydrate storage in stems[43,44] and by their dependence on carbohydrate reserves to regrow new xylem after hydraulic failure[45]. We, however, acknowledge that differences between angiosperms and gymnosperms could also influence the slower recovery in the latter, including their typically evergreen leaf habit (all gymnosperms included in our study were evergreen, whereas all angiosperms except *Nothofagus dombeyi* were deciduous)[46]. Further research is needed to identify the mechanisms underlying drought legacy effects in gymnosperm growth.

**Soil but not aridity and size affect resilience to drought.** Despite the large variation in climate across our study sites

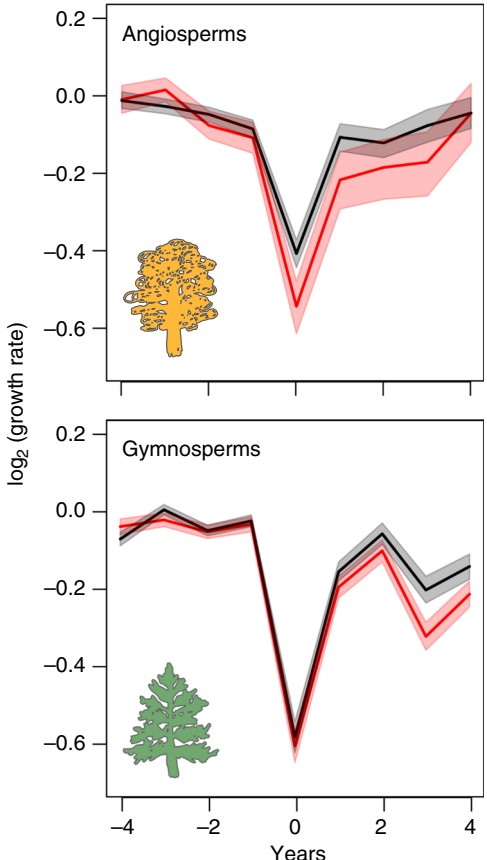

**Fig. 3 Growth patterns before, during and after the drought event studied (year = 0) for angiosperms and gymnosperms.** Data are presented as the average of log ratio between tree-ring width (TRW) at a given year and the average growth for the 4-year pre-drought period for surviving (black lines) and now-dead (red lines) trees. Shaded areas represent the 95% confidence intervals of the means from bootstrapping (1000 resamplings). Source data are available in Digital.CSIC repository (https://doi.org/10.20350/digitalCSIC/10536).

(Fig. 1), we do not find consistent effects of aridity on growth resilience and its components. On the one hand, we find a significant positive interaction between tree status and aridity on resilience and, to a lower extent, on resistance (Table 1, Supplementary Fig. 6), suggesting that surviving trees in more humid sites are more resilient due to a better capacity to resist the initial drought impact. On the other hand, recovery is always independent from aridity, and aridity has no effect on resistance when BAI is used as a measure of growth instead of TRW (Supplementary Table 3). Recent large-scale studies reported that resistance increases with the ratio between soil moisture and PET, whereas resilience increases with precipitation[34,47]. Results concerning the recovery appear mixed, with slower[30] or faster[34,48] recovery in dry than in wet forests. These contrasting findings may indicate that vulnerability to drought is to some extent decoupled from the exposure to climatic water deficits precisely because species distributions reflect (and tend to adjust to) the long-term climatic environment. Consistent with this view, hydraulic safety margins have been shown to be similar across biomes regardless of their rainfall environment[39]. Moreover, differences between studies could be explained by different biotic and abiotic conditions, experimental designs and modelling approaches. Overall, a universal pattern of the relationship between local climate and resilience remains hitherto inconclusive and deserves further investigation.

It is noteworthy that resilience and its components are also independent of the intensity of the drought (measured as the absolute Standardised Precipitation Evapotranspiration Index (SPEI)[49] or the SPEI difference between the relevant periods; see "Methods" and Table 1 and Supplementary Tables 2–4). This is consistent with a recent global study that did not find a strong link between the magnitude of the legacy effects after drought and its intensity[30].

Soil properties influence growth responses to drought. Soil fertility increases resistance but reduces recovery, particularly in surviving trees. These opposing effects determine a neutral effect of soil fertility in the resilience of now-dead trees and a negative effect on the resilience of surviving trees (Table 1, Supplementary Fig. 6). The latter effect is consistent with a detrimental role of high nutrient availability on drought survival due to preferential biomass allocation aboveground[50]. It is also important to consider that drought can have direct effects on soil fertility[51] and on the composition of soil bacterial and fungal communities[52], which may underlie some of the legacy effects on tree growth reported in this study and elsewhere[30].

Finally, resilience but not its components is negatively related to tree size (diameter at breast height (DBH); Table 1, Supplementary Fig. 6). Similarly, several studies reported higher sensitivity to drought in larger trees, particularly in the tropics[53,54], probably due to hydraulic limitations related to tree height[55]. However, note that tree size and age effects may be confounded and cannot be disentangled in purely observational studies such as ours[56].

## Discussion
Even though previous studies have related tree growth patterns to drought-induced mortality both at local and global scales[17–22], our study provides, to our knowledge, the first empirical evidence linking low growth resilience to past droughts (sensu Lloret et al.[33]) with increased risk of tree mortality across species and regions. We acknowledge that the spatial coverage of our data set is limited largely to temperate, Mediterranean and boreal biomes, although it encompasses substantial variation in geographic and climatic conditions within these regions (Fig. 1). Lack of information from the tropics reflects current tree-ring data availability[57]. Expanding this type of analysis to tropical forests should be possible in the near future due to the renewed interest and methodological developments in tropical dendroecology[58].

We also show that the ability to resist the immediate impacts of drought is linked to long-term mortality risk in angiosperms, whereas recovery capacity appears to control the likelihood of drought-induced mortality in gymnosperms. Our results confirm that growth resilience to past and current droughts should be considered as a promising proxy to assess future mortality risk at the individual tree level, bringing new tools to identify early signals of mortality and improving our capacity to forecast forest die-off under future climates.

## Methods
**Growth data.** We selected TRW (mm) data sets from the pancontinental database compiled by Cailleret et al.[22], for which (i) both dying and surviving trees growing together at the same site were cored, (ii) all individual TRW series had been successfully cross-dated and (iii) mortality was mainly induced by drought, solely or in combination with other factors, such as bark beetles, fungi or mistletoes[6] (Fig. 1, Supplementary Data 1). The database included 127 sites mostly located in the boreal, temperate and Mediterranean biomes of North America and Europe. We used two metrics of tree growth, TRW and BAI (mm$^2$). Although TRW and BAI show size and age effects, BAI reduces the pure geometric effect associated with the increase in tree DBH.

**Drought characterisation.** To explore the effect of drought on tree growth, we used the SPEI as a drought metric[49]. The SPEI is a multi-scale drought index calculated from the monthly difference between precipitation and PET. The

specific SPEI index we employed uses precipitation data from the Climate Research Unit (CRU TS 3.22) data set and PET data from the FAO-56 Penman–Monteith estimation, with a spatial resolution of 0.5°. Positive and negative SPEI values correspond to relatively wet and dry conditions, respectively. The SPEI can be computed at different time scales to characterise the duration and intensity of droughts because the SPEI value assigned to a particular month is calculated based on the averaged SPEI values of a time window covering the previous $n$ months. For each study site, SPEI values from 1901 to 2013 with a 1- to 24-month time windows were obtained from SPEIbase v.2.3 (2014, http://hdl.handle.net/10261/104742). The SPEI has been used for large-scale and long-term studies because it allows the comparison among sites with contrasting climates and accounts for timing and duration of drought during the current and previous years[59].

**Selecting the SPEI time windows**. Extreme drought events were detected by combining information from growth and SPEI time series. We selected the optimal time window of SPEI that maximised the goodness-of-fit of a linear model between SPEI and the residual chronology of standardised growth for each site. We considered different SPEI time windows for each site because species and biomes respond to drought at specific time scales[24,59]. First, we standardised the TRW series with the R package "dplR"[60] in the R environment[61]. Each individual TRW series was fit to a cubic smoothing spline (the frequency response was set to 0.50 at a wavelength of 67% of the time span of each series). An autoregressive model was then applied to the individual fitted series and the residuals of the resulting model were standardised by dividing them by the mean. Finally, a residual chronology for each site was obtained by averaging the individual residual indices using Tukey's biweight robust mean. This standardisation was flexible enough to preserve high-frequency climatic information while removing the low-frequency variability caused by tree age or size, or by external disturbances[62,63].

Second, we used monthly SPEI values for 24 different month scales (from 1 month to 2 years), ending in 5 target months corresponding to summer and early autumn (from June to October for the sites located in the Northern Hemisphere and from December of the previous year to May of the current year of ring growth in the Southern Hemisphere). For each site, we analysed the relationship between the residual chronology and the 120 different SPEI windows (5 target months × 24 month scales) for the common period (1931–1980) using linear regression models. For each of the 120 linear models per site, residual chronologies were the response variable and the SPEI for a given time window was the fixed explanatory variable. A Gaussian error distribution with an identity link was used. We compared model performance using the Akaike Information Criterion (AIC) to select the best model for each site, and its corresponding SPEI time window was used for the subsequent analyses (see Supplementary Data 2).

**Detecting extreme drought events**. We selected a single drought event for each site because the frequency and intensity of drought events can differ among sites[49]. The drought event was selected within a 30-year period following two steps. First, to discard exceptional long mortality periods, we excluded now-dead trees that died >50 years before the last death event recorded at a given site. Second, to avoid selecting drought events either too close or too distant in time to the mortality event, we limited the time period from 10 to 40 years before the first tree dying in each site (Supplementary Fig. 1). Within this period, we selected one drought event per site using two criteria: (1) SPEI less than the 10th percentile of the site-specific SPEI distribution (Supplementary Fig. 1a), and (2) abnormal low growth in the same year or in the year after (mean TRW of the site was reduced >5% relative to the average TRW of the 4 previous years). Growth reductions the year after the drought were rare but were considered, because depending on drought timing and the species tolerance, some trees might show a delay in their growth response to drought[64] (e.g. Supplementary Fig. 1b). After discarding 9 sites that did not meet the latter criteria, we considered 118 sites with 2456 and 1454 co-occurring, surviving and now-dead trees, respectively, of 22 species (8 angiosperms and 14 gymnosperms) for the subsequent analyses (Supplementary Fig. 1, Supplementary Data 1 and Data 2).

**Computing indices of resistance, recovery and resilience**. We computed resistance, recovery and resilience indices as proposed by Lloret et al.[33], considering 4 years before and after the drought event in agreement with Anderegg et al.[30]. Other periods around the studied drought (from 1- to 8-year period) were also analysed resulting in similar outcomes (Supplementary Fig. 3).

The resilience indices were computed as:

$$\text{Resistance} = \text{Dr}/\text{PreDr} \tag{1}$$

$$\text{Recovery} = \text{PostDr}/\text{Dr} \tag{2}$$

$$\text{Resilience} = \text{PostDr}/\text{PreDr} = \text{resistance} \times \text{recovery} \tag{3}$$

where PreDr was defined as mean raw TRW of the preceding 4-year period, Dr as raw TRW of the drought year and PostDr as mean raw TRW of the subsequent 4-year period. All indices were also calculated using BAI as a measure of growth.

**Climate covariate**. To account for climate diversity found across the study populations, we used the Global Aridity Index that provides high-resolution (30 arc-seconds) raster climate data for the 1970–2000 period[65]. The Global Aridity Index is the ratio between mean annual precipitation and mean annual reference evapotranspiration, based upon the implementation of the Penman Monteith Evapotranspiration equation for reference crops and using the WorldClim 2.0 data (http://worldclim.org/version2). The aridity index indicates rainfall over potential vegetation water demand (aggregated on an annual basis), and its value thus increases under more humid conditions and decreases with more arid conditions.

**Soil covariate**. We accounted for the variation in soil properties by means of two high-resolution soil databases from the International Soil Reference and Information Centre (ISRIC–World Soil Information). First, we used the WISE30sec database (WISE Soil Property Databases) that comprises a set of harmonised soil profiles at 30 arc seconds (~1 km at the equator), including 20 soil properties derived from statistical analyses of ca. 21,000 soil profiles (https://www.isric.org/explore/wise-databases)[66]. Second, we used SoilGrids, a global gridded soil information system at 250 m that provides global predictions for standard numeric soil properties at seven standard depths (0, 5, 15, 30, 60, 100 and 200 cm) and depth to bedrock based on ca. 150,000 soil profiles (https://www.isric.org/explore/soilgrids)[67]. We included ten soil characteristics to describe soil fertility in the study populations: organic carbon (g kg⁻¹), total nitrogen (g kg⁻¹), carbon/nitrogen ratio, bulk density (kg dm⁻³) and available water capacity (from −33 to −1500 kPa; cm m⁻¹) from WISE30sec database and absolute depth to bedrock (cm), pH measured at 200 cm and clay, silt and sand content (%) measured at 60 cm from the SoilGrids250m database. We conducted an ordination analysis of these soil characteristics at the study sites using principal component analysis (PCA) with the "FactoMineR" R package[68] in the R environment[61]. The first component of this PCA explained 55% of the variance and was positively associated with nitrogen concentration, organic content and available water capacity (Supplementary Fig. 5). This component was interpreted as indicator of soil fertility and included as an explanatory variable in our statistical models.

**Comparison between surviving and now-dead trees**. We analysed growth differences between coexisting now-dead and surviving trees for the three different response variables (resistance, recovery and resilience) around a severe drought previous to the mortality event, using linear mixed models (LMMs). Resistance, recovery and resilience were log-transformed to satisfy normality of the LMM residuals and considered as the response variables assuming a Gaussian error distribution with an identity link. In the initial full LMM, the included fixed effects were: (1) tree status (surviving vs. now-dead), (2) taxonomic group (angiosperms vs. gymnosperms), (3) DBH in the year of the drought event (DBH$_i$, where $i$ refers to the target drought event); (4) the relative intensity of the drought event, expressed as the SPEI value during the drought event (SPEI$_i$) and the SPEI difference corresponding to each metric, for resistance,

$$\text{SPEIdiff}_{\text{resist}} = \text{SPEI}_i - \text{PreSPEI} \tag{4}$$

for recovery,

$$\text{SPEIdiff}_{\text{recov}} = \text{PostSPEI} - \text{SPEI}_i \tag{5}$$

and for resilience,

$$\text{SPEIdiff}_{\text{resil}} = \text{PostSPEI} - \text{PreSPEI} \tag{6}$$

(5) the length of the time period between the drought event and the last year recorded in the ring-width series of each tree (Δtime), to control for temporal effects (mortality risk might be more related to a drought that occurred 10 years ago than 40 years ago); (6) the average ratio between precipitation and PET as a measure of climatic aridity; (7) the first principal component of the soil PCA as a measure of soil fertility; and (8) all the interactions between tree status (surviving vs. now-dead) and each of the other fixed effects. Because of the strong collinearity between SPEI$_i$ and SPEIdiff$_{\text{resist}}$ ($R^2 = 0.88$, $P < 0.001$, $N = 5928$) and between SPEI$_i$ and SPEIdiff$_{\text{recov}}$ ($R^2 = -0.89$, $P < 0.001$, $N = 5928$), we only considered the effect of both SPEI$_i$ and SPEIdiff$_{\text{resil}}$ for the resilience LMM (SPEIdiff$_{\text{resil}}$ vs. SPEI$_i$; $R^2 = -0.24$, $P > 0.05$, $N = 5928$). In all LMMs, random effects were estimated for the intercept with site nested in species and species nested in genus as grouping factors. We simplified each full LMM (Supplementary Table 1) by removing the least significant terms until a minimum adequate model (in terms of AIC) was identified[69] (see Supplementary Table 2 for model selection).

Identical analyses were also performed for resilience indices based on BAI instead of TRW (Supplementary Tables 3–5; Supplementary Fig. 2). In all cases, data exploration, model fitting, variance analyses and pairwise differences between the effects were computed with the R library "HighstatLib"[70] and the R packages "lme4"[71], "car"[72], "emmeans"[73] and "effects"[74] in the R environment[61].

**Additional models**. We tested the effects of the source of mortality and the interaction between taxonomic group and aridity on resilience indices based on TRW. We used additional models separately from the main models for the sake of clarity.

We fitted models including the additional source of mortality (Supplementary Data 1) as reported in the original studies (cf. Cailleret et al.[22]) as fixed effect (Supplementary Tables 6–8). These models did not perform better compared to the selected models reported in Table 1 and Supplementary Table 1 (see Supplementary Table 2 for model comparison), and hence this variable was not included in our final models.

We fitted models including the interaction between taxonomic group and aridity index to account for a potential covariation between these two variables. This interaction was never significant, and the AIC of the corresponding models was always slightly higher than that of the full model. We therefore did not include the interaction between taxonomic group and aridity index in the final models (see Supplementary Tables 2 and 9).

Note that all additional models were first compared with the full models (Supplementary Table 1) and then with reduced models (Table 1) when the effect of the newly added variables was significant.

**Reporting summary**. Further information on research design is available in the Nature Research Reporting Summary linked to this article.

## Data availability

The data that support this study are available in the Plant Trait database (TRY), https://www.try-db.org/. The source data of Figs. 2 and 3 are available in Digital.CSIC repository (https://doi.org/10.20350/digitalCSIC/10536).

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

## Acknowledgements

This article is based upon work from the COST Action FP1106 STReESS, financially supported by European Cooperation in Science and Technology (COST). L.DS. was funded by the Fundação para a Ciência e a Tecnologia (SFRH/BPD/70632/2010) and by the European Union (EU) under a Marie Skłodowska-Curie IF (No.797188); K.K. was supported by the Dutch Ministry of Agriculture, Nature and Food-quality (KB-29-009-003); E.M.R.R. by the Research Foundation – Flanders (FWO, Belgium) and by the EU under a Marie Skłodowska-Curie IF (No.659191); T.A. by the Kone Foundation; J.J.C. by the Spanish Ministry of Science (CGL2015-69186-C2-1-R); K.C. by the Slovenian Research Agency ARRS (P4-0015); L.J.H. by the USDA Forest Service-Forest Health Protection and Arkansas Agricultural Experiment Station; V.I.K. by the RFBR (18-45-240003 and 18-05-00432); T. Klein by the Merle S. Cahn Foundation and the Monroe and Marjorie Burk Fund for Alternative Energy Studies (Mr. and Mrs. Norman Reiser), the Weizmann Center for New Scientists and the Edith & Nathan Goldberg Career Development Chair; T.L. by the Slovene Research Agency (P4-0107, J4-5519 and J4-8216); J.C.L. by the Spanish Ministry of Science (CGL2013-48843-C2-2-R); H.M. by the Academy of Finland (No.315495); G.S.-B. by a Juan de la Cierva-Formación from the Spanish Ministry of Economy and Competitiveness (MINECO, FJCI 2016-30121); D.B.S. by the Ministry of Education and Science of the Republic of Serbia (III 43007); R.V. partially by BNP-PARIBAS Foundation; and J.M.-V. by the MINECO (CGL2013-46808-R and CGL2017-89149-C2-1-R) and an ICREA Academia award. Finally, we specially thank M. Berdugo, V. Granda, J. Moya, R. Poyatos, L. Santos del Blanco and R. Torices for their assistance in R programming.

## Author contributions

L.DS., J.M.-V., M.C., F.S., S.J., K.K. and E.M.R.R. conceived the ideas and designed the methodology. M.C., T.A., M.M.A., C.B., J.J.C., K.Č., G.G.-I., S.G., L.J.H., A.-M.H., J.M.K., V.I.K., T. Kitzberger, T. Klein, T.L., J.C.L., H.M., W.O., A.P., B.R., G.S.-B., D.B.S., M.L.S., R.V., and J.M.-V. collected the tree-ring data. L.DS., M.C., S.J., E.M.R.R. and J.M.-V. compiled and cleaned the ring-width database. L.DS. analysed the data, drafted and led the writing of the manuscript with inputs from J.M.-V., M.C., F.S., S.J., K.K. and E.M.R.R. All authors contributed critically to the drafts and gave final approval for publication.

## Competing interests

The authors declare no competing interests.

## Additional information

Lucía DeSoto[1,2]*, Maxime Cailleret[3,4,5], Frank Sterck[6], Steven Jansen[7], Koen Kramer[6,8], Elisabeth M.R. Robert[9,10,11], Tuomas Aakala[12], Mariano M. Amoroso[13], Christof Bigler[4], J. Julio Camarero[14], Katarina Čufar[15], Guillermo Gea-Izquierdo[16], Sten Gillner[17], Laurel J. Haavik[18], Ana-Maria Hereș[19,20], Jeffrey M. Kane[21], Vyacheslav I. Kharuk[22,23], Thomas Kitzberger[24,25], Tamir Klein[26], Tom Levanič[27], Juan C. Linares[28], Harri Mäkinen[29], Walter Oberhuber[30], Andreas Papadopoulos[31], Brigitte Rohner[5,4], Gabriel Sangüesa-Barreda[32], Dejan B. Stojanovic[33], Maria Laura Suárez[34], Ricardo Villalba[35] & Jordi Martínez-Vilalta[9,36]

[1]Estación Experimental de Zonas Áridas, Spanish National Research Council (EEZA-CSIC), Almería, Spain. [2]Centre for Functional Ecology, University of Coimbra, Coimbra, Portugal. [3]INRAE, Université Aix-Marseille, UMR Recover, Aix-en-Provence, France. [4]Forest Ecology, Department of Environmental Systems Science, ETH Zürich, Zürich, Switzerland. [5]Swiss Federal Institute for Forest, Snow and Landscape Research (WSL), Birmensdorf, Switzerland. [6]Forest Ecology and Forest Management Group, Wageningen University, Wageningen, The Netherlands. [7]Institute of Systematic Botany and Ecology, Ulm University, Ulm, Germany. [8]Land Life Company, Amsterdam, Netherlands. [9]CREAF, Bellaterrra (Cerdanyola del Vallès), Catalonia, Spain. [10]Ecology and Biodiversity, Vrije Universiteit Brussel, Brussels, Belgium. [11]Laboratory of Wood Biology and Xylarium, Royal Museum for Central Africa (RMCA), Tervuren, Belgium. [12]Department of Forest Sciences, University of Helsinki, Helsinki, Finland. [13]Instituto de Investigaciones en Recursos Naturales, Agroecología y Desarrollo Rural (IRNAD), Universidad Nacional de Río Negro, Consejo Nacional de Investigaciones Científicas y Técnicas (CONICET), Río Negro, Argentina. [14]Instituto Pirenaico de Ecología, Spanish National Research Council (IPE-CSIC), Zaragoza, Spain. [15]Department of Wood Science and Technology, Biotechnical Faculty, University of Ljubljana, Ljubljana, Slovenia. [16]Centro de Investigación Forestal (CIFOR), Instituto Nacional de Investigación y Tecnología Agraria y Alimentaria (INIA), Madrid, Spain. [17]Institute of Forest Botany and Forest Zoology, TU Dresden, Dresden, Germany. [18]USDA Forest Service, Missoula, MT, USA. [19]Department of Forest Sciences, Transilvania University of Brasov, Brasov, Romania. [20]BC3 - Basque Centre for Climate Change, Leioa, Spain. [21]Department of Forestry and Wildland Resources, Humboldt State University, Arcata, CA, USA. [22]Sukachev Institute of Forest, Siberian Division of the Russian Academy of Sciences (RAS), Krasnoyarsk, Russia. [23]Siberian Federal University, Krasnoyarsk, Russia. [24]Instituto de Investigaciones en Biodiversidad y Medio Ambiente (INIBOMA), Consejo Nacional de Investigaciones Científicas y Técnicas (CONICET), Bariloche, Argentina. [25]Department of Ecology, Universidad Nacional del Comahue, Río Negro, Argentina. [26]Department of Plant & Environmental Sciences, Weizmann Institute of Science, Rehovot, Israel. [27]Department of Yield and Silviculture, Slovenian Forestry Institute, Ljubljana, Slovenia. [28]Department of Physical, Chemical and Natural Systems, Pablo de Olavide University, Seville, Spain. [29]Natural Resources Institute Finland (Luke), Espoo, Finland. [30]Department of Botany, University of Innsbruck, Innsbruck, Austria. [31]Agricultural University of Athens, Karpenissi, Greece. [32]EiFAB-iuFOR, University of Valladolid, Soria, Spain. [33]Institute of Lowland Forestry and Environment, University of Novi Sad, Novi Sad, Serbia. [34]Grupo Ecología Forestal, CONICET - INTA, EEA Bariloche, Bariloche, Argentina. [35]Instituto Argentino de Nivología Glaciología y Ciencias Ambientales (IANIGLA-CONICET), Mendoza, Argentina. [36]Universitat Autònoma de Barcelona, Bellaterrra (Cerdanyola del Vallès), Catalonia, Spain. *email: luciadesoto@gmail.com

