## [Peer Review File · Nature Communications]

Reviewers' comments:

Reviewer #1 (Remarks to the Author):

This paper aims at establishing a link between resilience components of past drought events based on tree-ring series and mortality due to drought. The authors used a database of 3500 trees from 118 sites from Europe, Asian Russia, the USA and Argentina/Chile. At each site, both surviving and dead trees have been sampled. At about half of the sites, drought was identified as the sole reason for mortality, and on the other half as a contributing factor. Using a precipitation/evaporation index, and pointer years in the tree-ring series, one drought event was selected on each site, and the resilience components according to Lloret et al. 2011 were calculated. These were then compared with a mixed model, where the following factors were considered: Survival, taxa (Angiosperms and Gymnosperms), diameter at breast height in the drought year, time between the drought year and the last recorded tree ring and interactions between survival and genus and survival and diameter at breast height.

The authors found a significant effect of survival on resilience. This effect was different for the two taxa: Surviving angiosperm trees showed higher resistance than not-surviving angiosperm trees. No difference in resistance was found for gymnosperms. However, surviving gymnosperms had lower recovery than not-surviving gymnosperms. The authors confirm that using the resilience components according to Lloret et al. 2011 is a valid method for assessing susceptibility for drought-induced mortality and that there are important differences between taxa. For angiosperms, both low resilience and resistance is an indicator for future mortality, while for gymnosperms, low resilience and low recovery is an indicator for mortality.

The drought indicators as defined by Lloret et al. are based on the idea that lower growth precedes mortality in trees. This theory is based on many studies, e.g. Pedersen 1998 and more recently Ogle et al. 2000, Liang et al. 2003, Biegler and Burgmann 2003, Das et al. 2007, Biegler et al. 2006 & 2007. The drought indicators by Lloret are used widely in climate change impact studies on trees and therefore a direct link to mortality is interesting for the field.

The strength of the paper is that dead and living trees have been sampled on the same site, allowing for a comparison of factors that contributed to the mortality. I'm therefore enthusiastic about the question asked and think that the idea of looking at different responses from angiosperms and gymnosperms would be fundamentally insightful to understand what causes trees to succumb to drought and why. The idea of linking the Lloret indicators to mortality is important and of great concern to many scientists.

Despite this initial enthusiasm I have some concerns that the current analysis may not be appropriate for the claims made in the title and in the manuscript, and that more evidence should be provided that confounding effects do not drive the main results of this paper.

1. The main drawback of this study is that angiosperms and gymnosperms have not been observed on the same site. It seems to be assumed that the differences found were due to the taxonomic group and not due to spatial effects. The authors did include a precipitation-evaporation index in the analysis. This may control for some differences in climate (resolution of the climate data is low), yet still leaving the effect of soil unaccounted for.

2. The phrasing of the title and in the paper suggests that the authors tried to predict mortality from past resilience to drought, which is not exactly what the statistical analysis is about. Because different effects on resistance, resilience and recovery are evaluated, the authors do more of an association analysis. A better way to analyze this dataset would be to use mortality as the response variable in a binomial generalized mixed model and have the components of resilience and the additional variables (taxa, P/PET etc.) with their interactions as fixed effects. This would allow for a more detailed analysis and would allow to quantify the probability of death due to low resistance/resilience.

3. While the dataset that the authors used is impressive considering the condition that dead and surviving trees had to be present, the spatial coverage does not seem to be very balanced. I wonder how this might influence the results and how confident we can be that these results can be generalized as the paper implies.

4. In its current form, the paper is very short and would benefit from more detail, for example: Results for the individual species could be shown in a figure.

5. It would benefit from additional figures that explain the dataset better, e.g. the confounding effects of dbh, P/PET, age etc. Maybe in the supplementary information.

6. Was the interaction between group and P/PET tested? This might also help to control for site differences.

7. The aspect of plasticity is not discussed in the paper. Trees have mechanisms to adjust to changing environmental conditions and lower resilience may also indicate a structural adjustment to drier conditions.

8. Confounding effect of age: Tree age can have an effect on resilience components (Lloret et al. 2011). While the authors included an effect for tree size (DBH), this may not always be related to age across species.

9. The source of mortality should also be included in the model, as it may explain the differences between taxa.

10. Figure 2 & 3: Please back-transform the data. It is not possible to assess the meaningfulness of the differences on a log scale.

Further points:

Figure 2: Is this figure really showing SE or rather CI?

Line 110-113: I am also not aware of a study that looked at mortality as a function of Lloret's resilience indicators. However, there are many studies that have linked the underlying growth patterns to mortality (see below). Therefore, the results are not completely novel as the same has been described with different approaches.

Line 128: Maybe rephrase to: Around the globe

Line 245-246: This is very speculative, and this paper does not provide any evidence related to hydraulic failure.

Line 255-256: It may also indicate that these trees shed foliage and reduce fine root mass. For example: Nilsson, L. O., & Wiklund, K. (1992). Influence of nutrient and water stress on Norway spruce production in south Sweden—the role of air pollutants. *Plant and Soil*, 147(2), 251-265.

Line 261: Unclear what the relevance of resprouting is in this context. Please remove or explain further.

Line 265: This should be discussed in more detail or removed. What role might different leaf habits play in relation to mortality due to drought?

Line 266-268: The lack of an effect here may also be related to the low resolution of the climate data used (.5° ≈ 55km).

Line 295-297: I think this should be rephrased to include a reference to the resilience components as defined by Lloret et al 2011, as there are several studies that have linked growth patterns during drought to mortality.

Line 555: Discount may not be the right word here, as it implies a mathematical adjustment.

Table S1: Change "Wood bores" to Wood borer

References:

Bigler, Christof, et al. "Drought induces lagged tree mortality in a subalpine forest in the Rocky Mountains." *Oikos* 116.12 (2007): 1983-1994.

Bigler, C., Bräker, O. U., Bugmann, H., Dobbertin, M., & Rigling, A. (2006). Drought as an inciting mortality factor in Scots pine stands of the Valais, Switzerland. *Ecosystems*, 9(3), 330-343.

Bigler, C., & Bugmann, H. (2003). Growth-dependent tree mortality models based on tree rings. *Canadian Journal of Forest Research*, 33(2), 210-221.

Das, A. J., Battles, J. J., Stephenson, N. L., & Van Mantgem, P. J. (2007). The relationship between tree growth patterns and likelihood of mortality: a study of two tree species in the Sierra Nevada. *Canadian Journal of Forest Research*, 37(3), 580-597.

Liang, E., Shao, X., Kong, Z., & Lin, J. (2003). The extreme drought in the 1920s and its effect on tree growth deduced from tree ring analysis: a case study in North China. *Annals of Forest Science*, 60(2), 145-152.

Ogle, Kiona, Thomas G. Whitham, and Neil S. Cobb. "Tree-ring variation in pinyon predicts likelihood of death following severe drought." *Ecology* 81.11 (2000): 3237-3243.

Pedersen, B. S. (1998). The role of stress in the mortality of midwestern oaks as indicated by growth prior to death. *Ecology*, 79(1), 79-93.

Reviewer #2 (Remarks to the Author):

I enjoyed reading paper by DeSoto et al. It is well-written analysis looking at the effects of resilience, resistance, and recovery to drought using tree rings, gridded CRU climate data, and mixed effects models. The work is timely and important in the fields of ecology and will likely appeal to a broad audience of ecologists. However, the novelty is somewhat diminished because Gazol et al (2017) (cited in this text), perform a very similar analysis of resistance, resilience, and recovery using the ITRDB database.

Other concerns besides the novelty include the fact that the authors use only one drought metric and one climate product. Though CRU and SPEI are a good first pass, it would also be useful to try reanalysis products such as the Sheffield et al. 0.25 climate forcing (which would also increase the author's sample size given that it is at a higher resolution).

Secondly, all of the analyses seem to rely on the drought lasting 1 year and 4-year, post and pre-drought periods. I would like to see sensitivity tests to this window length, periods before the drought. Also, it seems like if the authors' model selected the 24-month SPEI time window, it might bias the calculation of resilience and resistance (because the 4-year pre drought window would potentially include some drought years). Thus, the assumption of a 1 year drought period seems a bit problematic.

Methodological issues are all fixable given a revision.

Line-specific comment:

L108 abrupt transition. I suggest starting a new paragraph

Reply to reviewers' comments

Reviewer #1 (Remarks to the Author):

This paper aims at establishing a link between resilience components of past drought events based on tree-ring series and mortality due to drought. The authors The idea of linking the Lloret indicators to mortality is important and of great concern to many scientists.

Despite this initial enthusiasm I have some concerns that the current analysis may not be appropriate for the claims made in the title and in the manuscript, and that more evidence should be provided that confounding effects do not drive the main results of this paper.

1. The main drawback of this study is that angiosperms and gymnosperms have not been observed on the same site. It seems to be assumed that the differences found were due to the taxonomic group and not due to spatial effects. The authors did include a precipitation-evaporation index in the analysis. This may control for some differences in climate (resolution of the climate data is low), yet still leaving the effect of soil unaccounted for.

Regarding the latter concern, we agree with the referee and, therefore, we added a new variable coding for soil fertility to our models, based on a high-resolution gridded dataset of soil properties (see Methods lines 662-682). Although this soil variable was significant in many of our models (see results in Table 1, S2, S4-S5; and new text describing these results in lines 298-303), our main conclusions regarding the relationship between resilience and mortality and the differences between taxonomic groups remain unchanged.

Regarding the former, we understand the referee's concern about the lack of cooccurrences of angiosperms and gymnosperms at the same sites. This reflects to a large extent the fact that these two plant groups do not coexist in many areas and, where they do, our study is constrained by the tree-ring data available. However, most angiosperms and gymnosperms are present in the same biomes (see Fig. 1b) and their sampling locations substantially overlap in terms of latitude and longitude (see figure). The histograms represents the number of trees in our study (green bars, gymnosperms; orange bars, angiosperms) by 20 degrees (~2000km) of latitude and longitude.

Nevertheless, we checked the importance of the spatial effect with two different approximations. First, we tested the significance of the effect of latitude and longitude by adding them as fixed effects in our models (see supplementary material, Table S8). These models showed that the effects of latitude and longitude were never significant and always resulted in poorer model fit ($\Delta\text{AIC} \gg 2.0$ relative to the same model without these variables). This remained true regardless of whether the effect of the taxonomic group was included or excluded from the model (Tables S8, S9). Second, we tested the spatial autocorrelation of our data by explicitly modelling random effects with a spatial correlation structure. Again, these alternative models showed higher AICs ($\Delta\text{AIC} \gg 2.0$ relative to the same model without spatial random structure) (Tables S7, S8). Overall, these results indicate that spatial effects do not seem to be very important in our study and, most importantly, that our main results are robust to these spatial effects (see new text in lines 720-730 in main text).

2. The phrasing of the title and in the paper suggests that the authors tried to predict mortality from past resilience to drought, which is not exactly what the statistical analysis is about. Because different effects on resistance, resilience and recovery are evaluated, the authors do more of an association analysis. A better way to analyze this dataset would be to use mortality as the response variable in a binomial generalized mixed model and have the components of resilience and the additional variables (taxa, P/PET etc.) with their interactions as fixed effects. This would allow for a more detailed analysis and would allow to quantify the probability of death due to low resistance/resilience.

We agree with the referee that we do not directly test the effect of drought resilience (and its components) on tree mortality, but rather our analysis examines the association between these two phenomena. The referee suggests testing mortality as the response variable in a binomial generalized mixed model with components of resilience (and the additional variables) as fixed effects. We had considered this approach when designing the current study (and in fact used it in a previous study, Cailleret et al. 2016 Ecological Applications) and we have considered it again after receiving the reviewers' comments. However, we believe that in this case predicting resilience (or its components) from mortality is more appropriate to the sampling design and the data available than predicting mortality from resilience as suggested by the reviewer. The reasons are the following.

Firstly, tree status (living vs dead) was an *a priori* factor driving the sampling design of all the studies that contributed to our database (which does not necessarily reflect the real proportion of dead vs living trees in the sampled populations), whereas resilience and its components were estimated *a posteriori* and hence it seems more logical to consider these latter variables as the response variables. Secondly, and perhaps more important, resilience and its components were estimated for a given year, which was the same for all trees within a site, whereas mortality occurred at different times for different trees within a site, sometimes spanning decades. This makes the modelling of resilience and its components easier (as we can estimate explanatory variables such as SPEI or DBH for the relevant year/period for each site, which would be impossible for mortality) and more statistically sound. Therefore, we have decided to keep the models as they were, but modified the text to make it clearer that our results are

indicative of association and not an evidence of causality (see lines abstract, 313-317, and title).

3. While the dataset that the authors used is impressive considering the condition that dead and surviving trees had to be present, the spatial coverage does not seem to be very balanced. I wonder how this might influence the results and how confident we can be that these results can be generalized as the paper implies.

Our analysis is constrained by the available studies of tree mortality and radial growth, which are focused on extra-tropical forests (largely due to methodological constraints) and mainly in the Northern Hemisphere (this is a common issue in large-scale tree-ring studies; see Zhao et al. 2019, *J Biogeogr.* 46:355-368; or Babst et al. 2017, *Nature Ecol and Evol* 1:0008). Nevertheless, we included several populations of *Austrocedrus* and *Nothofagus* in the Southern Hemisphere and we covered vast areas in the Northern hemisphere between 31 and 63 degrees North and -112 and 93 East, and we included 22 species from 10 genera. Most importantly, our database included tree population with a climatic water balance in terms of Aridity Index ranging between 0.14 and 1.92, from arid to humid climates (these values are now included in Table S7). We also provide a new panel in Figure 1 (Figure 1b) including the location of our sampling sites in biome space, which shows that our sites were distributed across five of the main nine biomes (all except tundra, tropical rainforest, tropical forest savanna, and temperate rainforest). We therefore think that the spatial coverage of our analysis is quite extensive and offers a reasonable coverage of the world's forests outside the tropics. However, we have added new text highlighting the constraints of our database, which largely reflect the methodological constraints of dendrochronology as a discipline (lines 317-323). Finally, we have replaced the term 'global' in the text by 'pan-continental' to reflect the fact that data coverage was not strictly global, and also for consistency with other papers using the same database (Cailleret et al. 2017, *Global Change Biol* 23:1675-1690).

4. In its current form, the paper is very short and would benefit from more detail, for example: Results for the individual species could be shown in a figure.

We showed results by species in a new supplementary figure (Fig. S4) and refer to it in the main text (lines 176-178). We did not include this as result in the main paper because we wanted to keep it as short and clear as possible focusing on general trends, and discussing individual species' responses is clearly beyond the scope of our study and may suffer from the low sample size for some species.

5. It would benefit from additional figures that explain the dataset better, e.g. the confounding effects of dbh, P/PET, age etc. Maybe in the supplementary information.

We added a new figure in the supplementary information showing the effect of all the covariates used in the model (DBH, SPEI, aridity and soil) on resilience and its components, separately for angiosperms and gymnosperms (Fig. S6).

6. Was the interaction between group and P/PET tested? This might also help to control for site differences.

We acknowledge the suggestion of the referee. Therefore, we fitted the models again including the interaction between taxonomic group and aridity index. The effect of this interaction was not significant in any of the models, and hence was not included in the final models (see lines 736-740 and supplementary material, Table S13). Moreover, the AIC of these new models were typically higher and never significantly lower from the AIC of the corresponding full models (Table S3).

7. The aspect of plasticity is not discussed in the paper. Trees have mechanisms to adjust to changing environmental conditions and lower resilience may also indicate a structural adjustment to drier conditions.

We partially agree with the referee. Low resilience may reflect structural adjustment to drier conditions in some cases. However, the fact that structural adjustments are frequently mediated by changes in growth rates makes it difficult in our case to disentangle structural adjustments from direct (deleterious) drought effects. The conceptual figure below shows four hypothetical trees with identical pre-drought growth rates. Trees a and d (in orange) are more resilient than b and c (in purple), but trees a and b show lower plasticity in growth than trees c and d. Low resilience in trees b and c may reflect some sort of structural adjustment, but it is impossible to separate this from purely deleterious drought effects with the data available. In fact our results finding of lower resilience in now-dead trees (Figure 2a) clearly indicate the limits of plasticity. We now discuss briefly this aspect of the results in the text (lines 182-193).

8. Confounding effect of age: Tree age can have an effect on resilience components (Lloret et al. 2011). While the authors included an effect for tree size (DBH), this may not always be related to age across species.

We thank the reviewer for pointing this out. There are three main reasons for not including the effect of age explicitly in our models. First, age was not known for all cases, as not all sampled trees were cored through the pith. Second, age and size effects in trees are hard to disentangle in observational studies, as these two variables are closely related both within and among species. Third, there is consistent evidence from controlled, experimental studies showing that the apparent effect of age on trees is largely due to size (Mencuccini et al. 2005, *Ecol. letters* 8:1183-1190). A sentence has

been added to the main text indicating that size and age effects may be confounded (lines 310-312).

9. The source of mortality should also be included in the model, as it may explain the differences between taxa.

Please note that we only included datasets in which drought was identified by the original authors as the main factor causing the mortality event (see Methods), so the source of mortality is always drought or, at least, drought-related. It is also true that in some datasets additional sources of mortality were identified in association to drought. However, these sources are very unequally distributed among taxonomic groups (see table below), which makes its analysis difficult.

Furthermore, we tested the effect of the additional source of mortality by fitting two models in which this variable was included as explanatory factor in addition to or instead of taxonomic group (see Tables 10-12). These models did not perform better in terms of fit than those including taxonomic group, and hence this variable was not included in the final models (see Tables S3 and new text in lines 731-735).

additional source	taxa group	No. trees	No. sites
none	angiosperms	286	8
	gymnosperms	1353	67
Bb	angiosperms	0	0
	gymnosperms	568	20
Bb & F	angiosperms	0	0
	gymnosperms	88	2
C	angiosperms	0	0
	gymnosperms	152	4
F	angiosperms	96	2
	gymnosperms	452	12
M	angiosperms	0	0
	gymnosperms	256	5
Wb	angiosperms	813	7
	gymnosperms	0	0

Bb, bark beetles; C, competition; F, fungi; M, mistletoe; Wb, wood-borers.

10. Figure 2 & 3: Please back-transform the data. It is not possible to assess the meaningfulness of the differences on a log scale.

We back-transformed data in Figure 2. We did not back-transform the data in Figure 3 because the logarithms allow for a correct comparison of ratios. Log response ratios are a standard way of quantifying biological responses due to their desirable statistical properties (see, for instance, Hedges et al. 1999 *Ecology* 80:1150-1156). This is because, a double expression ratio is described by a log ratio of 1, a halved expression

rate can be described by a log ratio of -1 , and an unchanged expression rate can be described by a log ratio of zero.

Further points:

11. Figure 2: Is this figure really showing SE or rather CI?

It was SE. We changed it to CI in the new figure.

12. Line 110-113: I am also not aware of a study that looked at mortality as a function of Lloret's resilience indicators. However, there are many studies that have linked the underlying growth patterns to mortality (see below). Therefore, the results are not completely novel as the same has been described with different approaches.

We introduced the studies highlighted by the referee (lines 111-115).

13. Line 128: Maybe rephrase to: Around the globe

We have changed the text as suggested.

14. Line 245-246: This is very speculative, and this paper does not provide any evidence related to hydraulic failure.

We have removed this sentence.

*15. Line 255-256: It may also indicate that these trees shed foliage and reduce fine root mass. For example: Nilsson, L. O., & Wiklund, K. (1992). Influence of nutrient and water stress on Norway spruce production in south Sweden—the role of air pollutants. *Plant and Soil*, 147(2), 251-265.*

This idea has been incorporated into the text, in connection with possible structural adjustments (lines 184-193).

16. Line 261: Unclear what the relevance of resprouting is in this context. Please remove or explain further.

We have removed this sentence as suggested.

17. Line 265: This should be discussed in more detail or removed. What role might different leaf habits play in relation to mortality due to drought?

We have modified the text to make it clear that we just refer to additional factors related with taxonomic group (at least for the species included in our study) that could be influencing the differences we found between angiosperms and gymnosperms (Lines 267-271).

18. Line 266-268: The lack of an effect here may also be related to the low resolution of the climate data used (.5° = ~55km).

We now use higher-resolution climate data (see Methods, lines 652-661) and our main results remain unchanged. See second Reviewer #2's issue for a detailed explanation.

19. Line 295-297: I think this should be rephrased to include a reference to the resilience components as defined by Lloret et al 2011, as there are several studies that have linked growth patterns during drought to mortality.

To our knowledge no previous study has specifically explored the relationship between drought resilience in radial growth and drought-induced mortality. However, we have added a reference to Lloret et al. (2011) as requested to avoid any ambiguity regarding the type of resilience we are referring to.

20. Line 555: Discount may not be the right word here, as it implies a mathematical adjustment.

We changed the text as suggested.

21. Table S1: Change "Wood bores" to Wood borer

We corrected the text.

References:

- Bigler, Christof, et al. "Drought induces lagged tree mortality in a subalpine forest in the Rocky Mountains." *Oikos* 116.12 (2007): 1983-1994.
- Bigler, C., Bräker, O. U., Bugmann, H., Dobbertin, M., & Rigling, A. (2006). Drought as an inciting mortality factor in Scots pine stands of the Valais, Switzerland. *Ecosystems*, 9(3), 330-343.
- Bigler, C., & Bugmann, H. (2003). Growth-dependent tree mortality models based on tree rings. *Canadian Journal of Forest Research*, 33(2), 210-221.
- Das, A. J., Battles, J. J., Stephenson, N. L., & Van Mantgem, P. J. (2007). The relationship between tree growth patterns and likelihood of mortality: a study of two tree species in the Sierra Nevada. *Canadian Journal of Forest Research*, 37(3), 580-597.
- Liang, E., Shao, X., Kong, Z., & Lin, J. (2003). The extreme drought in the 1920s and its effect on tree growth deduced from tree ring analysis: a case study in North China.

Annals of Forest Science, 60(2), 145-152.

Ogle, Kiona, Thomas G. Whitham, and Neil S. Cobb. "Tree-ring variation in pinyon predicts likelihood of death following severe drought." *Ecology* 81.11 (2000): 3237-3243.

Pedersen, B. S. (1998). *The role of stress in the mortality of midwestern oaks as indicated by growth prior to death.* *Ecology*, 79(1), 79-93.

Reviewer #2 (Remarks to the Author):

1. I enjoyed reading paper by DeSoto et al. It is well-written analysis looking at the effects of resilience, resistance, and recovery to drought using tree rings, gridded CRU climate data, and mixed effects models. The work is timely and important in the fields of ecology and will likely appeal to a broad audience of ecologists. However, the novelty is somewhat diminished because Gazol et al (2017) (cited in this text), perform a very similar analysis of resistance, resilience, and recovery using the ITRDB database.

We agree with the referee that several previous studies have addressed the patterns and implications of resilience in radial growth. However, none of those to our knowledge has studied the association between resilience and tree mortality, which is the main focus of our contribution. For instance, the study by Gazol et al. (2017) cited by the reviewer does not contain any data on tree mortality nor on radial growth of dead trees. Therefore, we are still convinced that our study provides novel insights on the use of resilience indices as indicators of tree mortality risk that are of wide interest to forest ecologists.

2. Other concerns besides the novelty include the fact that the authors use only one drought metric and one climate product. Though CRU and SPEI are a good first pass, it would also be useful to try reanalysis products such as the Sheffield et al. 0.25 climate forcing (which would also increase the author's sample size given that it is at a higher resolution).

We agree with the referee that higher climatic resolution could provide more accurate results, and we reanalysed our data accordingly. Regarding the P/PET factor using CRU data in our former models, we changed it by the Global Aridity Index, a high-resolution (30 arc-seconds) raster climate dataset for the 1970-2000 period computed as the ratio between mean annual precipitation and mean annual reference evapotranspiration. For drought intensity characterization, we decided to keep using the SPEI database. The SPEI has a long time-span but low spatial resolution, with interpolated data ranging from 1901 to present with 0.5 degree resolution (corresponding to roughly 50x50 km cells). The SPEI is becoming a standard drought index for studies at wide spatial scales, including global analyses of tree growth patterns (e.g. Vicente-Serrano et al. 2013, PNAS 110:52-57), and has several advantages relative to alternative metrics (e.g., it accounts for differences in atmospheric water demand, it can be computed at different time scales) that makes it

particularly suitable for our study. So, we believe it is the most suitable index to characterize duration and intensity of droughts.

Please note that the Sheffield et al. 2006, J Clim 19:3088-3111 database (ranging from 1948 to 2016) does not cover the time span needed for 32 of our study sites (more than a quarter of the total populations studied) and hence could not be used in our case.

3. Secondly, all of the analyses seem to rely on the drought lasting 1 year and 4-year, post and pre-drought periods. I would like to see sensitivity tests to this window length, periods before the drought. Also, it seems like if the authors' model selected the 24-month SPEI time window, it might bias the calculation of resilience and resistance (because the 4-year pre drought window would potentially include some drought years). Thus, the assumption of a 1 year drought period seems a bit problematic.

Regarding the reviewers' first concern, we did explore the growth patterns of surviving and now-dead trees at different periods between 1 and 8 years before and after the target drought years. These results are reported in Figure S3, both in the previous and in the current version of the manuscript, and show that (1) the effect of drought is largely circumscribed to the drought year, and (2) the main patterns would not be affected by a different choice of the length of the pre- and post-drought periods (up to eight years), as growth patterns tend to be relatively stable within these periods. Additional analyses also showed that very low SPEI values during the pre- and post-drought periods were exceptional (see figure below). In this Figure, we show mean SPEI values before, during and after the drought event studied (year=0) for the studying sites. Sites with angiosperms or gymnosperms are shown separately. Shaded areas represent the 95% confidence intervals of the means from bootstrapping (1,000 resamplings).

Note also that we used different SPEI time-windows for each site, selected among 120 different SPEI windows (5 target months \times 24 month-scales). The selected SPEI window was > 12 months (1 year) for only 15% of the sites (Table S7).

4. Methodological issues are all fixable given a revision.

We addressed all the methodological issues raised by the reviewer.

5. Line-specific comment:

L108 abrupt transition. I suggest starting a new paragraph

We started a new paragraph as suggested.

REVIEWERS' COMMENTS:

Reviewer #1 (Remarks to the Author):

The authors have revised their manuscript with care and I am satisfied with the rebuttal to my concerns.

As some time has passed since this manuscript has been written, an update to the cited literature would improve the timeliness of the paper. For example, Isaac-Renton et al. analyze the physiological mechanisms behind resilience components after Lloret et al, which may be relevant for this paper <https://www.nature.com/articles/s41467-018-07701-0>.

The manuscript should be thoroughly checked for typos and grammar, as I came across some issues. For example, in line 111.

Reviewer #2 (Remarks to the Author):

I really like the addition of the new soil fertility analysis to the manuscript and I think it adds to the paper. However, I am concerned about the new analysis that attempts to account for spatial autocorrelation. The authors don't go into detail about the methodology and they don't mention the process of diagnosing spatial autocorrelation using a test such as Moran's I. Did the authors to check to see if spatial autocorrelation is a concern in the first place? What function did the authors use to model spatial structure? Standard practice is to include the latitude and longitude coordinates of each plot in the regression and test several (linear, quadratic ratio, exponential, spherical and Gaussian) spatial correlation structures and select the most likely and parsimonious model using AIC (see the Dormann paper below). AIC, however cannot be used to determine whether or not it is appropriate to account for spatial autocorrelation (as the authors have done, as I understand it), the appropriate test is Moran's I or something analogous. I'd be surprised if spatial autocorrelation is an issue given the plot distributions shown in Figure 1, thus I'd just recommend removing this analysis because either the authors have (i) not described their methodology clearly enough for me to understand fully what they've done or (ii) they have not accounted for potential spatial autocorrelation correctly.

Dormann, C. F. et al. Methods to account for spatial autocorrelation in the analysis of species distributional data: a review. *Ecography* 30, 609–628 (2007).

Overall I find this to be a very interesting read, but it might be more appropriate for a more field specific journal given my previously voiced novelty concerns.

Reviewer #1 (Remarks to the Author):

The authors have revised their manuscript with care and I am satisfied with the rebuttal to my concerns.

As some time has passed since this manuscript has been written, an update to the cited literature would improve the timeliness of the paper. For example, Isaac-Renton et al. analyze the physiological mechanisms behind resilience components after Lloret et al, which may be relevant for this paper <https://www.nature.com/articles/s41467-018-07701-0>.

We acknowledge the suggestion of the Reviewer and included it.

The manuscript should be thoroughly checked for typos and grammar, as I came across some issues. For example, in line 111.

We checked the manuscript for typos and grammar mistakes.

Reviewer #2 (Remarks to the Author):

I really like the addition of the new soil fertility analysis to the manuscript and I think it adds to the paper. However, I am concerned about the new analysis that attempts to account for spatial autocorrelation. The authors don't go into detail about the methodology and they don't mention the process of diagnosing spatial autocorrelation using a test such as Moran's I. Did the authors to check to see if spatial autocorrelation is a concern in the first place? What function did the authors use to model spatial structure? Standard practice is to include the latitude and longitude coordinates of each plot in the regression and test several (linear, quadratic ratio, exponential, spherical and Gaussian) spatial correlation structures and select the most likely and parsimonious model using AIC (see the Dormann paper below). AIC, however cannot be used to determine whether or not it is appropriate to account for spatial autocorrelation (as the authors have done, as I understand it), the appropriate test is Moran's I or something analogous. I'd be surprised if spatial autocorrelation is an issue given the plot distributions shown in Figure 1, thus I'd just recommend removing this analysis because either the authors have (i) not described

their methodology clearly enough for me to understand fully what they've done or (ii) they have not accounted for potential spatial autocorrelation correctly.

Dormann, C. F. et al. Methods to account for spatial autocorrelation in the analysis of species distributional data: a review. *Ecography* 30, 609–628 (2007).

Overall I find this to be a very interesting read, but it might be more appropriate for a more field specific journal given my previously voiced novelty concerns.

We decided to remove the analyses accounting for spatial autocorrelation as suggested by the Reviewer and the Editor because: (1) the spatial distribution of sampling sites indeed suggests that spatial autocorrelation effects are unlikely, (2) our own analyses indicate that these effects are not significant in our models, and (3) the number and complexity of the models reported in the Supplementary Information is already quite high and we feel that simplifying the additional analyses by removing the spatial models may make the paper clearer and more accessible to a wide readership.